# URLOST: Unsupervised Representation Learning without Stationarity or Topology

## Abstract

Unsupervised representation learning has seen tremendous progress but is constrained by its reliance on data modality-specific stationarity and topology, a limitation not found in biological intelligence systems. For instance, human vision processes visual signals derived from irregular and non-stationary sampling lattices yet accurately perceives the geometry of the world. We introduce a novel framework that learns from high-dimensional data lacking stationarity and topology. Our model combines a learnable self-organizing layer, density adjusted spectral clustering, and masked autoencoders. We evaluate its effectiveness on simulated biological vision data, neural recordings from the primary visual cortex, and gene expression datasets. Compared to state-of-the-art unsupervised learning methods like SimCLR and MAE, our model excels at learning meaningful representations across diverse modalities without depending on stationarity or topology. It also outperforms other methods not dependent on these factors, setting a new benchmark in the field. This work represents a step toward unsupervised learning methods that can generalize across diverse high-dimensional data modalities.

## 1 Introduction

Unsupervised representation learning, also known as self-supervised representation learning (SSL), aims to develop models that autonomously detect patterns in data and make these patterns readily apparent through a specific representation. There has been tremendous progress over the past few years in the unsupervised representation learning community. Popular methods like contrastive learning and masked autoencoders [68; 6; 24; 70] work relatively well on typical modalities such as images, videos, audio, time series, and point clouds. However, these methods make implicit assumptions about the data domain's **topology** and **stationarity**. Given an image, **topology** refers to the neighboring pixels of each pixel, or more generally, the grid structure in images, the temporal structure in time series and sequences, or the 3D structure in molecules and point clouds. **Stationarity** refers to the property that the low-level statistics of the signal remain consistent across its domain. For instance, pixels and patches in images exhibit similar low-level statistics (mean, variance, co-variance) regardless of their locations within the domain. The success of state-of-the-art self-supervised representation learning relies on knowing the prior topology and stationarity of the modalities. For example, joint-embedding SSL employs random-resized cropping augmentation [6], and masked auto-encoding [25] utilizes masked-image-patch augmentation. What if we possess high-dimensional signals without knowledge of their domain topology or stationarity? Can we still craft a high-quality representation? This is not only the situation that biological vision systems have to deal with but also a practical setting for many scientific data analysis problems. In this work, we introduce *unsupervised representation learning without stationarity or topology* (**URLOST**) and take a step in this direction.

As we mentioned earlier, typical modalities possess topology and stationarity prior information that can be utilized by unsupervised representation learning. Taking images as an example, digital cameras employ a consistent sensor grid that spans the entire visual field. However, biological visual systems have to deal with signals with less domain regularity. For instance, unlike camera sensors which have a uniform grid, the cones and rods in the retina distribute unevenly and non-uniformly. This results in a non-stationary raw signal. ~~Retinal ganglion cells connect to more photoreceptors in the fovea than in the periphery. The correlation of the visual signal between two different locations in the retina depends not only on the displacement between these locations but~~

~~also on their absolute positions.~~ Yet, biological visual systems can establish precise retinotopy from the retina to neurons based on spontaneous locally-propagated retinal activities and external stimuli [67; 35; 18] and leverage retinotopic input to build unsupervised representation. This implies that we can potentially build unsupervised representation without relying on prior stationarity of the raw signal or topology of the input domain. The ability to build unsupervised representation without relying on topology and stationarity has huge advantages. In evolution perspective, biological visual system develops irregular sensor for a reason. It gives us foveated vision which has both high resolution and broad coverage. Such sensor arrangement is much more efficient than camera sensor for dynamic environments. In order to use these efficient sensors, our visual system must be able to develop an unsupervised learning model that does not rely on topology and stationarity of the signal. On the other hand, if we can build such a model, we can potentially create an embodied vision system with more efficient sensors than standard camera sensor. We can even move beyond visual signal and create a powerful AI system that compute with any high dimensional signal.

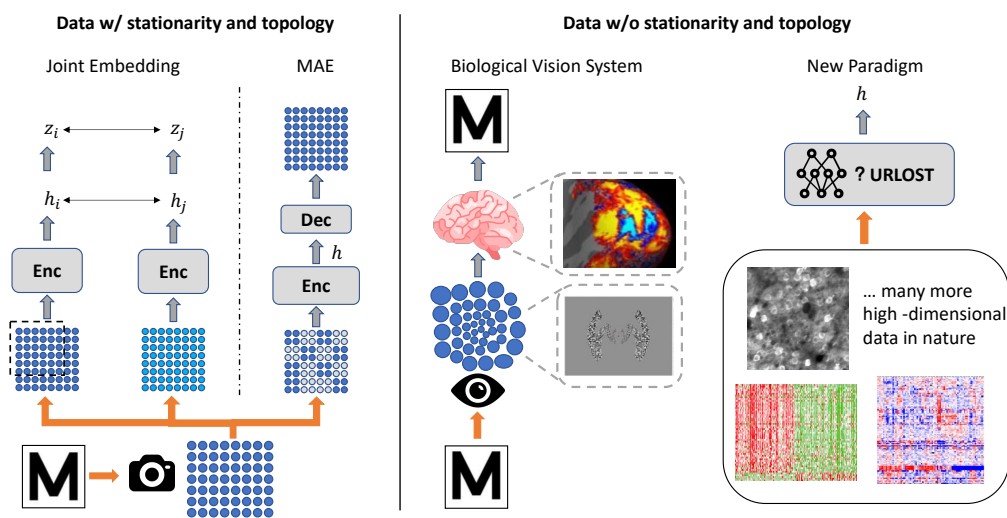

Figure 1: From left to right: the unsupervised representation learning through joint embedding and masked auto-encoding; the biological vision system that perceives via unstructured sensor and understands signal without stationarity or topology [48]; and many more such diverse high dimensional signal in natural science that our method supports while most existing unsupervised methods don't. Data figures are borrowed from [48; 44; 69].

In this work, we aim to build unsupervised representations for general high-dimensional vectors. Taking images as an example again, let's assume we receive a set of images whose pixels are shuffled in the same order. How can we build representations in an unsupervised fashion without knowledge of the shuffling order? If possible, can we use such a method to build unsupervised representations for general high-dimensional data? Inspired by [53], we use low-level statistics and spectral clustering to form clusters of the pixels, which recovers a coarse topology of the input domain. These clusters are analogous to image patches except that they are slightly irregularly shaped and different in size. We mask a proportion of these "patches" and utilize a locally connected neural network and and a Vision Transformer [15] to predict the masked "patches" based on the remaining unmasked ones. This "learning to predict masked tokens" approach is proposed in masked autoencoders (MAE) [25] and has demonstrated effectiveness on typical modalities. Initially, we test the proposed method on the synthesized biological visual dataset, derived from CIFAR-10 [32] using a foveated retinal sampling mechanism [8]. Then we generalize this method to two high-dimensional vector datasets: a primary visual cortex neural response decoding dataset [57] and the TCGA miRNA-based cancer classification dataset [62; 66]. Across all these benchmarks, our proposed method outperforms existing SSL techniques, establishing its effectiveness in building unsupervised representations for signals lacking explicit stationarity or topology. Given the emergence of new modalities in deep learning from natural sciences [59; 23; 49; 34], such as chemistry, biology, and neuroscience, our

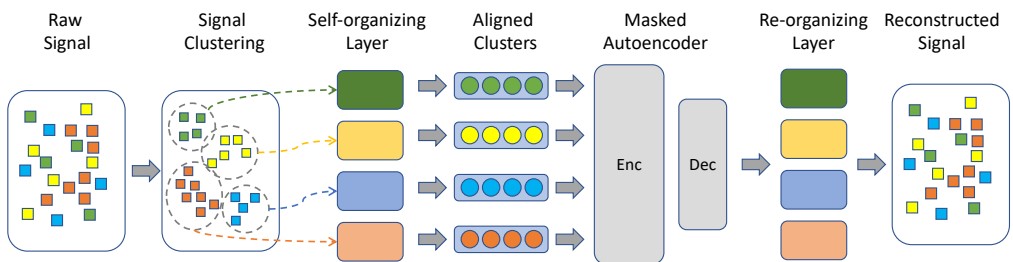

Figure 2: **The overview framework of URLOST.** The high-dimensional input signal undergoes clustering and self-organization before unsupervised learning using a masked autoencoder for signal reconstruction.

method offers a promising approach in the effort to build unsupervised representations for high-dimensional data.

Main contribution of this work:

1. We identify an important self-supervised learning problem that is largely ignored by the machine learning community: how to build unsupervised representation for general high-dimension data? High-dimensional data is prevalent in everyday life, scientific research, and nature.

2. And, we propose a straightforward yet effective method that provides a step for solving this problem, which combines intuition for high dimensional statistics, bio-inspired design and state-or-the-art self-supervised learning method. Specifically, our model is inspired by formation of retinotopy and how visual systems compute with retinotopic input.

3. We show our model can deal with diverse modalities and have numerous applications. For example, it can serve as a vision foundation model with more efficient sensor arrangement. It can also be used as a hypothesis testing tool on neural data and a cancer diagnosis tool.

## 2 METHOD

### 2.1 MOTIVATION AND OVERALL FRAMEWORK

Our objective is to build robust, unsupervised representations for high-dimensional signals that lack explicit topology and stationarity. These learned representations are intended to enhance performance in downstream tasks, such as classification. To achieve this, we begin by using low-level statistics and clustering to approximate the signal's topology. The clusters derived from the signal serve as input to a masked autoencoder. As depicted in Figure 1, the masked autoencoder randomly masks out patches in an image and trains a Transformer-based autoencoder unsupervisedly to reconstruct the original image. After the unsupervised training, the autoencoder's latent state yields high-quality representations. In our approach, signal clusters are input to the masked autoencoder.

Notably, the clusters differ from image patches in several key aspects due to the differences in the input signal: they are unaligned, exhibit varied sizes and shapes, and their clustering nodes are not confined to fixed 2D locations like pixels in image patches. To cope with these differences, we introduce a self-organizing layer responsible for aligning these clusters through learnable transformations. The parameters of this layer are jointly optimized with those of the masked autoencoder. Our method is termed URLOST, an acronym for **U**nsupervised **R**epresentation **L**earning with**O**ut **S**tationarity or **T**opology. Figure 2 provides an overview of the framework. URLOST consists of three core components: density adjusted spectral clustering, self-organizing layer, and masked autoencoder. The functionalities of these components are detailed in the following subsections.

## 2.2 DENSITY ADJUSTED SPECTRAL CLUSTERING

Representation learning for high-dimensional signals without explicit topology is challenging. We propose to define a metric to measure inter-dimensional relationships. This metric effectively approximates a topology for the signal. Similar to [53], where they use the absolute correlation values as the metric for pixels, we employ a more general metric: discrete mutual information. (refer to Appendix A.2) as the metric. Let affinity matrix $A_{ij}$ denote the mutual information between dimension $i$ and $j$, which approximates the manifold $\mathcal{M}$ that the signal lives on. We can define the discretized Laplacian operator based on $A$ and use the eigenvector of the Laplacian operator to perform spectral clustering, which segments the manifold. The detailed definition and the algorithm are left in Appendix A.2. Finding the eigenvector of the Laplacian operator is a discretized approximation of the following optimization problem in function space: $\min_{\|f\|_{L^2(\mathcal{M})}} \int_{\mathcal{M}} \|p(x)^{\frac{1}{2}} \nabla f(x)\|^2$ where $f(x) : \mathcal{M} \to [0,1]$ is the normalized signal defined on $\mathcal{M}$ and $p(x)$ is the density function. The integral is taken over standard measure on $\mathcal{M}$. Since spectral clustering heavily relies on the solution of equation 6, the definition of the density function $p(x)$ affects the quality of the resulting clusters. Given a high dimensional dataset $S \in \mathbb{R}^{n \times m}$, let $S_i \in \mathbb{R}^{n \times 1}$ be $i$th column of $S$, which represents the $i$th dimension of the $m$-dimensional signal. Since we want to process dimensions that share similar information together, we use spectral clustering to group them with mutual information graph. Let $A_{ij} = I(S_i; S_j)$, we use $A$ as the affinity matrix of the graph. $L = D - A$ is the Laplacian matrix, where D is the diagonal matrix whose $(i, i)$-element is the sum of $A$'s $i$-th row. We can formulate the spectral embedding problem as finding $Y$ such that $\min_{YY^T=I} tr(YLY^T)$. A clustering algorithm is then applied to the embedding $Y$ as explained in appendix A.2. The size and shape of the clusters strongly affect the unsupervised learning performance. To adjust the size and shape, we apply a density adjustment matrix $P$ to adjust $L$ in the spectral embedding objective. The optimization problem becomes the following:

$$\min_{YY^T=I} tr(YP^{1/2}LP^{1/2}Y^T) \tag{1}$$

where $P = diag(p(i))$, $p(i)$ is the density function defined on each node $i$. We set $p(i) = q(i)^\alpha n(i)^{-\beta}$[1], where $n(i) = \sum_{j \in \text{Top}_K(A_{ji})} A_{ji}$ and $q(i)$ is the prior density which depends on specific dataset and is defined in the experiment section. $\alpha$, $\beta$ and $K$ are hyper-parameters. Setting $\alpha = 0$ and $K = m$ will recover the normalized graph Laplacian. In appendix A.1, we provide a detailed interpretation and motivation of density adjustment with the language of functional analysis. We further verify its effectiveness with ablation study in section 4.2 and appendix A.4

## 2.3 SELF-ORGANIZING LAYER

Transforming a high-dimensional signal into a sequence of clusters using the above method is not enough because it does not capture the internal structure within individual clusters. As an intuitive example, given an image, we divide it into a set of image patches of the same size. If we apply different permutations to these image patches, their inner product will no longer reflect their similarity properly. Clusters we obtained from section 2.2 are analogous to image patches, but elements in each cluster have arbitrary ordering. Thus, if we take two clusters of pixels, their inner product are also arbitrary due to the ordering mismatch. In Transformers, since self-attention depends on the inner products between different "clusters," we need to align these clusters in a space, where their inner products reflect their similarity. To effectively perform unsupervised learning on these clusters, it is essential to align them in some manner. Directly solving the exact alignment problem with low-level statistics of the signal is challenging. To align these clustersThus, we propose a *self-organizing layer* with learnable parameters. Specifically, let vector $x^{(i)}$ denote the $i$th cluster. Each cluster $x^{(i)}$ is passed through a differentiable function $g(\cdot, w^{(i)})$ with parameter $w^{(i)}$, resulting in a sequence $z_0$:

$$z_0 = [g(x^{(1)}, w^{(1)}), \cdots g(x^{(M)}, w^{(M)})] \tag{2}$$

$z_0$ is comprised of projected and aligned representations for all clusters. The weights of the proposed self-organizing layer, $\{w^{(1)}, \cdots w^{(M)}\}$, are jointly optimized with the subsequent neural network introduced in the next subsection.

---

[1] $p(i)$ need to normalized to be the density function but this wouldn't influence the optimization problem

## 2.4 MASKED AUTOENCODER

After the self-organizing layer, $z_0$ is passed to a Transformer-based masked autoencoder (MAE) with an unsupervised learning objective. Masked autoencoder (MAE) consists of an encoder and a decoder which both consist of stacked Transformer blocks introduced in [64]. The objective function is introduced in [25]: masking random image patches in an image and training an autoencoder to reconstruct them, as illustrated in Figure 1. In our case, randomly selected clusters in $z_0$ are masked out, and the autoencoder is trained to reconstruct these masked clusters. After training, the encoder's output is treated as the learned representation of the input signal for downstream tasks. The masked prediction loss is computed as the mean square error (MSE) between the values of the masked clusters and their corresponding predictions.

## 3 RESULT

Since our method is inspired by the biological vision system, we first validate its ability on a synthetic biological vision dataset created from CIFAR-10. Then we evaluate the generalizability of URLOST on two high-dimensional natural datasets collected from diverse domains. Detailed information about each dataset and the corresponding experiments is presented in the following subsections. Across all tasks, URLOST consistently outperforms other strong unsupervised representation learning methods.

## 3.1 SYNTHETIC BIOLOGICAL VISION DATASET

As discussed in the introduction, the biological visual signal serves as an ideal dataset to validate the capability of URLOST. In contrast to digital images captured by a fixed array of sensors, the biological visual signal is acquired through irregularly positioned ganglion cells, inherently lacking explicit topology and stationarity. However, it is hard to collect real-world biological vision signals with high precision. Therefore, we employ a retinal sampling technique to modify the classic CIFAR-10 dataset and simulate imaging from the biological vision signal. The synthetic dataset is referred to as *Foveated CIFAR-10*. To make a comprehensive comparison, we also conduct experiments on the original CIFAR-10, and a *Permuted CIFAR-10* dataset obtained by randomly permuting the image.

**Permuted CIFAR-10.** To remove the grid topology inherent in digital imaging, we simply permute all the pixels within the image, which effectively discards any information related to the grid structure of the original digital image. We applied such permutation to each image in the CIFAR-10 dataset to generate the *Permuted CIFAR-10* dataset. Nevertheless, permuting pixels only removes an image's topology, leaving its stationarity intact. To obtain the synthetic biological vision that has neither topology nor stationarity, we introduce the *Foveated CIFAR-10*.

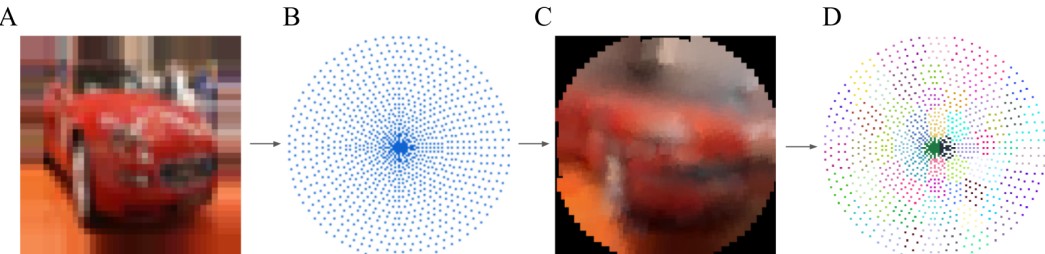

Figure 3: **Retina sampling** (A) An image in CIFAR-10 dataset. (B) Retina sampling lattice. Each blue dot represents the center of a Gaussian kernel, which mimics a retinal ganglion cell. (C) Visualization of the car image's signal sampled using the retina lattice. Each kernel's sampled RGB value is displayed at its respective lattice location for visualization purposes. (D) density-adjusted spectral clustering results are shown. Each unique color represents a cluster, with each kernel colored according to its assigned cluster.

**Foveated CIFAR-10.** Much like photosensors installed in a camera, retina ganglion cells within the primate biological visual system sample from visual stimuli and project images. However, unlike

Table 1: **Evaluation on computer vision and synthetic biological vision dataset.** ViT (Patch) stands for the Vision Transformer backbone with image patches as inputs. ViT (Pixel) means pixels are treated as input units. ViT (Clusters) means clusters are treated as inputs instead of patches. The number of clusters is set to 64 for both *Permuted CIFAR-10* and *Foveated CIFAR-10* dataset. Eval Acc stands for linear probing evaluation accuracy: accuracy of a linear classifier on our pretrained model.

| Dataset | Method | Backbone | Eval Acc |
|---|---|---|---|
| CIFAR-10 | MAE | ViT (Patch) | **88.3 %** |
| | MAE | ViT (Pixel) | 56.7 % |
| | SimCLR | ResNet-18 | 90.7 % |
| Permuted CIFAR-10 (no topology) | URLOST MAE | ViT (Cluster) | **86.4 %** |
| | MAE | ViT (Pixel) | 56.7 % |
| | SimCLR | ResNet-18 | 47.9 % |
| Foveated CIFAR-10 (no topology or stationarity) | URLOST MAE | ViT (Cluster) | **85.4 %** |
| | MAE | ViT (Pixel) | 48.5 % |
| | SimCLR | ResNet-18 | 38.0 % |

photosensors that have uniform receptive fields and adhere to a consistent sampling pattern, retinal ganglion cells at different locations of the retina vary in their receptive field size: smaller in the center (fovea) but larger in the peripheral of the retina. This distinctive retina sampling pattern results in foveated imaging [63]. It gives primates the ability to have both a high-resolution vision and a broad overall receptive field while consequently making visual signals sampled by the retina lack stationarity. The evidence is that responses of two ganglion cells separated by the same displacement are highly correlated in the retina but less correlated in the peripheral. To mimic the foveated imaging with CIFAR-10, we adopt the retina sampling mechanism from [8]. Specifically, each retina ganglion cell is simplified and modeled using a Gaussian kernel. The response of each cell is determined by the dot product between pixel values and the Gaussian kernel. Figure 3 illustrates the sampling kernel locations. Applying this sampling grid and permuting the resulting pixels produces the *foveated CIFAR-10*. In the natural retina, retinal ganglion cell density decreases linearly with eccentricity, which makes fovea much denser than the peripheral, compared to the simulated lattice in Figure 3. However, considering the low resolution of the CIFAR-10 dataset, we reduce the simulated fovea's density to prevent redundant sampling. In this dataset, we define the prior density $q(i)$ as the distance from $i$th sampling kernel to the center of the sampling lattice.

**Experiments.** We compare URLOST on both of the synthetic vision datasets as well as the original CIFAR-10 with popular unsupervised representation learning methods SimCLR [6] and MAE [25]. We follow the standard evaluation framework for unsupervised learning model. All ~~the~~ models conducted unsupervised learning followed by linear probing ~~for classification accuracy.~~ i.e. we first train our unsupervised learning model on the data, which is generally known as the pretraining phase. Then we convert each data point to an embedding vector with our pretrained encoder. Finally, we perform logistic regression on embedding vectors with their labels and record the accuracy. The evaluations are reported in Table 1. SimCLR excels on CIFAR-10 but struggles badly with both synthetic datasets due to its inability to handle data without stationarity and topology. MAE gets close to SimCLR on CIFAR-10 with a $4 \times 4$ patch size. However, the patch size no longer makes sense when data has no topology. So we additionally tested MAE masking pixels instead of image patches. It maintains the same performance on *Permuted CIFAR-10* as on CIFAR-10, though poorly, invariant to the removal of topology as it should be. But It still drops greatly to $48.5\%$ on the *Foveated CIFAR-10* when stationarity is also removed. In contrast, only URLOST is able to maintain consistently strong performances when there is no topology or stationarity, achieving $86.4\%$ on *Permuted CIFAR-10* and $85.4\%$ on *Foveated CIFAR-10* when the baselines completely fail. For the best result, we use $K = 20$, $\alpha = 0.5$ and $\beta = 2$. Further ablation study on the effect of these hyper-parameter is discussed in the ablation study 4.2.

Table 2: **Evaluation on V1 response decoding and TCGA pan-cancer classification tasks.** "Raw" indicates preprocessed (standardized and normalized) raw signals. Best $\beta$ values are used for $\beta$-VAE. For URLOST MAE, cluster sizes are 200 (V1) and 32 (TCGA). We pick 15 seeds randomly to repeat the training and evaluation for each method. We report the 95% confidence interval for each method.

| Method | V1 Response Decoding Acc | TCGA Classification Acc |
|---|---|---|
| Raw | 73.9% | 91.1% |
| $\beta$-VAE | ~~75.1%~~ $75.64\% \pm 0.11\%$ | ~~94.2%~~ $94.15\% \pm 0.24\%$ |
| MAE | 64.8% | 88.3% |
| URLOST MAE | ~~**78.2%**~~ **78.75% ± 0.18 %** | ~~**94.9%**~~ **94.90% ± 0.25 %** |

## 3.2 V1 NEURAL RESPONSE TO NATURAL IMAGE STIMULUS

After accessing URLOST's performance on synthetic biological vision data, we take a step further to challenge its generalizability with high-dimensional natural datasets. The first task is decoding neural response recording in the primary visual area (V1) of mice.

**V1 neural response dataset.** The dataset, published by [44], contains responses from over 10,000 V1 neurons captured via two-photon calcium imaging. These neurons responded to 2,800 unique images from ImageNet [12], with each image presented twice to assess the consistency of the neural response. In the decoding task, a prediction is considered accurate if the neural response to a given stimulus in the first presentation closely matches the response to the same stimulus in the second presentation within the representation space. This task presents greater challenges than the synthetic biological vision described in the prior section. For one, the data comes from real-world neural recordings rather than a curated dataset like CIFAR-10. For another, the geometric structure of the V1 area is substantially more intricate than that of the retina. To date, no precise mathematical model of the V1 neural response has been well established. The inherent topology and stationarity of the data still remain difficult to grasp [43; 42]. Nevertheless, evidence of Retinotopy [18; 19] and findings from prior research [41; 7; 58] suggest that the neuron population code in V1 are tiling a low dimensional manifold. This insight led us to treat the population neuron response as high-dimensional data and explore whether URLOST can effectively learn its representation.

**Experiments.** Following the approach in [44] we apply standardization and normalization to the neural firing rate. The processed signals are high-dimensional vectors, and they can be directly used for the decoding task, which serves as the "raw" signal baseline in Table 2. For representation learning methods, URLOST is evaluated along with MAE and $\beta$-VAE [26]. Note that the baseline methods need to handle high-dimensional vector data without stationarity or topology, so SimCLR is no longer applicable. We use $\beta$-VAE instead. We first train the neural network with an unsupervised learning task, then use the latent state of the network as the representation for the neural responses in the decoding task. The results are presented in the table 2. Our method surpasses the original neuron response and other methods, achieving the best performance. For the density function, since we have no prior knowledge on the nodes in this dataset, we set $K = 15$, $\alpha = 0$ and $\beta = 1$

## 3.3 GENE EXPRESSION DATA

In this subsection, we further evaluate URLOST on high-dimensional natural science data from a completely different domain, the gene expression data.

**Gene expression dataset.** The dataset comes from The Cancer Genome Atlas (TCGA) [62; 66], which is a project that catalogs the genetic mutations responsible for cancer using genome sequencing and bioinformatics. The project molecularly characterized over 20,000 primary cancers and matched normal samples spanning 33 cancer types. We focus on the pan-cancer classification task: diagnose and classify the type of cancer for a given patient based on his gene expression profile. The TCGA project collects the data of 11,000 patients and uses Micro-RNA (miRNA) as their gene expression profiles. Like the V1 response, no explicit topology and stationarity are known and each data point is a high-dimensional vector. Specifically, 1773 miRNA identifiers are used so that each data point is a 1773-dimensional vector. Types of cancer that each patient is diagnosed with serve as the classification labels.

Table 3: **Ablation study on self-organizing layer**. Linear probing accuracy with varying parameters, keeping others constant. For *Locally-Permutated CIFAR-10*, we use $4 \times 4$ patch size. For *Permutated CIFAR-10* and *Foveated CIFAR-10*, we set the number of clusters to 64 for the spectral clustering algorithm. We kept the hyperparameter of the backbone model the same as in table 1.

| Dataset | Projection | Eval Acc |
|---|---|---|
| Locally-Permuted CIFAR-10 | shared | 81.4 % |
|  | non-shared | **87.6** % |
| Permuted CIFAR-10 | shared | 80.7 % |
|  | non-shared | **86.4** % |

(a) Replacing the non-shared projections of the self-organizing layer with the shared projection layer entails a significant drop in performance.

| Dataset | Cluster | Eval Acc |
|---|---|---|
| Foveated CIFAR-10 | SC | 82.7 % |
|  | DSC | **85.4** % |

(b) "SC" denotes spectral clustering with uniform density clustering and "DSC" denotes density adjusted spectral clustering. For *Foveated CIFAR-10* using density adjusted spectral clustering to create clusters will make the model perform better than using standard spectral clustering with uniform density.

**Experiments.** Similar to Section 3.2, URLOST is compared with the original signals, MAE, and $\beta$-VAE, which is the state-of-the-art unsupervised learning method on TCGA cancer classification [71; 72]. We also randomly partition the dataset do five-fold cross-validation and report the average performance in Table 2. Again, our method learns meaningful representation from the original signal. The learned representation benefited the classification task and achieved the best performance, demonstrating URLOST's ability to learn meaningful representation of data from diverse domains. For the density function, since we have no prior knowledge on the nodes in this dataset, we set $K = 15$, $\alpha = 0$ and $\beta = 1$.

## 4 ABLATION STUDY

### 4.1 SELF-ORGANIZING LAYER VS SHARED PROJECTION LAYER

Conventional SSL models take a sequential input $x = [x^{(1)}, \cdots x^{(M)}]$ and embed them into latent vectors with a linear transformation:

$$z_0 = [Ex^{(1)}, \cdots Ex^{(M)}] \tag{3}$$

which is further processed by a neural network. The sequential inputs can be a list of language tokens [14; 50], pixel values [5], image patches [15], or overlapped image patches [6; 24; 70]. $E$ can be considered as a projection layer that is shared among all elements in the input sequence. The self-organizing layer $g(\cdot, w^{(i)})$ introduced in Section 2.3 can be considered as a *non-shared projection layer*. We conducted an ablation study comparing the two designs to demonstrate the effectiveness of the self-organizing layers both quantitatively and qualitatively. To facilitate the ablation, we further synthesized another dataset.

**Locally-permuted CIFAR-10**. To directly evaluate the performance of the non-shared projection approach, we designed an experiment involving intentionally misaligned clusters. In this experiment, we divide each image into patches and locally permute all the patches. The $i$-th image patch is denoted by $x^{(i)}$, and its permuted version, permutated by the permutation matrix $E^{(i)}$, is expressed as $E^{(i)}x^{(i)}$. We refer to this manipulated dataset as the *Locally-Permuted CIFAR-10*. Our hypothesis posits that models using shared projections, as defined in Equation 3, will struggle to adapt to random permutations, whereas self-organizing layers equipped with non-shared projections can autonomously adapt to each patch's permutation, resulting in robust performance. This hypothesis is evaluated quantitatively and through the visualization of learned weights $w^{(i)}$.

**Permuted CIFAR-10**. Meanwhile, we also run the ablation study on the *Permuted CIFAR-10*. Unlike locally permuted CIFAR-10, a visualization check is not viable since the permutation is done globally. However, we can still quantitatively measure the performance of the task.

**Quantitative results**. Table 3 confirms our hypothesis, demonstrating a significant performance decline in models employing shared projections when exposed to permuted data. In contrast, the non-shared projection model maintains stable performance.

**Visual evidence**. Using linear layers to parameterize the self-organizing layers, i.e. let $g(x, W^{(i)}) = W^{(i)}x$, we expect that if the projection layer effectively aligns the input sequence, $E^{(i)T}W^{(i)}$ should

exhibit visual similarities. That is, after applying the inverse permutation $E^{(i)T}$, the learned projection matrix $W^{(i)}$ at each location should appear consistent or similar. The proof of this statement is provided in Appendix A.5. The model trained on *Locally-Permuted CIFAR10* provides visual evidence supporting this claim. In Figure 4, the weights show similar patterns after reversing the permutations. These observations demonstrate that URLOST can also be used as an unsupervised learning method to recover topology and enforce stationary on the signal. This is different than just using the model to extract representations for downstream tasks. We believe this is a unique potential of URLOST.

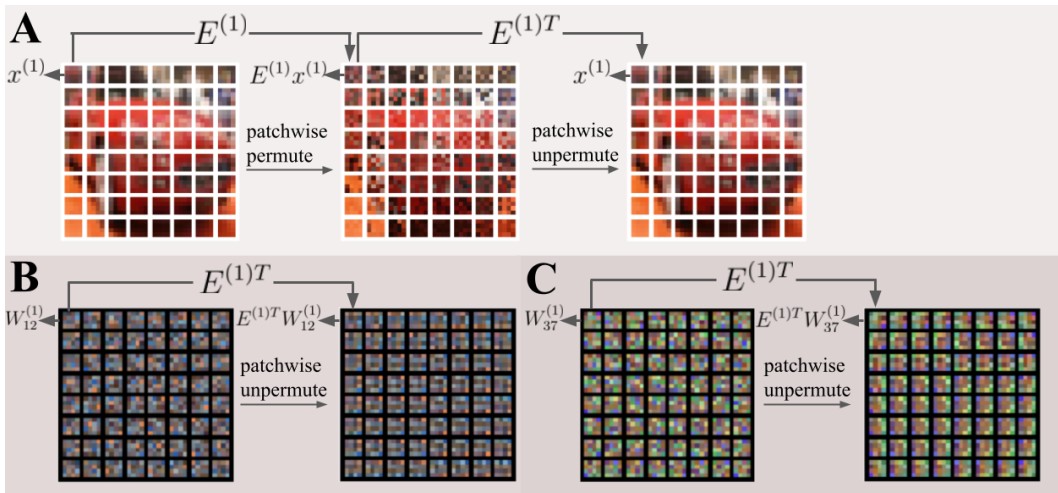

Figure 4: **Learnt weights of a self-organizing layer.** (A) Image is cropped into patches, where each patch $x^{(i)}$ first undergoes a different permutation $E^{(i)}$, then the inverse permutation $E^{(i)T}$. (B) The learned weight of the linear self-organizing layer. The 12th column of $W^{(i)}$ at all positions $i$ are reshaped into patches and visualized. When $W^{(i)}$ undergoes the inverse permutation $E^{(i)T}$, they show similar patterns. (C) Visualization of the 37th column of $W^{(i)}$. Similar to (B).

## 4.2 DENSITY ADJUSTED CLUSTERING VS UNIFORM DENSITY CLUSTERING

As explained in Section 2.2, the shape and size of each cluster depend on how the density function $p(i)$ is defined. ~~Let $q(i)$ represent the eccentricity, the distance from $i$th kernel to the center of the sampling lattice, and let $n(i) = \sum_j A_{ji}$ where $A$ is the affinity matrix, then the density is defined as:~~

$$p(i) = q(i)^\alpha n(i)^{-\beta} \tag{4}$$

where $n(i) = \sum_{j \in \text{Top}_k(A_{ji})} A_{ji}$, $A$ is the affinity matrix, $q(i)$ represent the eccentricity, the distance from $i$th kernel to the center of the sampling lattice. Setting $\alpha$ and $\beta$ nonzero, the density function is eccentricity-dependent. Setting both $\alpha$ and $\beta$ to zero will make $n(i)$ constant which recovers the uniform density spectral clustering. We vary the parameters $\alpha$ and $\beta$ to generate different sets of clusters for the foveated CIFAR-10 dataset and run URLOST using each of these sets of clusters. Results in Table 4 validate that the model performs better with density adjusted clustering. The intuitive explanation is that by adjusting the values of $\alpha$ and $\beta$, we can make each cluster carry similar amounts of information (refer to Appendix A.4.). A balanced distribution of information across clusters enhances the model's ability to learn meaningful representations. Without this balance, masking a low-information cluster makes the prediction task trivial, while masking a high-information cluster will make the prediction task too difficult. In either scenario, the model's ability to learn effective representations is compromised.

## 5 ADDITIONAL RELATED WORKS

Several interconnected pursuits are linked to this work, and we will briefly address them here:

**Topology in biological visual signal.** 2-D topology of natural images is strong prior that requires many bits to encode [13; 2]. Such 2-D topology is encoded in the natural image statistic [55; 28],

which can be recovered [53] and [31]. However, these "topology recovering" work cannot be effectively integrated with state-of-the-art self-supervised learning algorithms. For example, [53] use manifold learning to infer the 2-d position of each pixel. The community tried to feed the "recovered topology" to a graph neural network (GNN) [3], but suffer from inherent scalability issues on using GNN to do unsupervised learning. Optic and neural circuits in the retina result in a more irregular 2-D topology than the natural image, which can still be simulated [52; 46; 47; 45; 61; 30]. This information is further processed by the primary visual cortex. Evidence of retinotopy suggests the low-dimensional geometry of visual input from retina is encoded by the neuron in primary visual cortex [40; 20; 27; 19; 65; 48]. These evidences suggest we can recover the topology using signal from retinal ganglion cell and V1 neurons.

**Evidence of self-organizing mechanism in the brain.** In computational neuroscience, many works use the self-organizing maps (SOM) as a computational model for V1 functional organization: [16; 60; 1; 17; 39; 31]. In other words, this idea of self-organizing is likely a principle governing how the brain performs computations. Even though V1 functional organizations are present at birth, numerous studies also indicate that the brain's self-organizing mechanisms continue after full development [22; 54; 29].

**Learning with signal on non-euclidean geometry.** In recent years, researchers from the machine learning community have made efforts to consider geometries and special structures beyond classic images, text, and feature vectors. [33] treats an image as a set of points but depends on the 2D coordinates. The geometric deep learning community tries to generalize convolution neural networks beyond the Euclidean domain [3; 37; 11; 21]. Recent research also explores adapting the Transformer to domains beyond Euclidean spaces [10; 9]. However, none of them has tried to tackle the issue when the data has no explicit topology or stationarity, which is the focus of URLOST.

**Self-supervised learning.** Self-supervised learning (SSL) has made substantial progress in recent years. Different SSL method is designed for each modality, for example: predicting the masked/next token in NLP[14; 50; 4], solving pre-text tasks, predicting masked patches, or building contrastive image pairs in computer vision [36; 25; 68; 6; 24; 70]. These SSL methods have demonstrated descent scalability with a vast amount of unlabeled data and have shown their power by achieving performance on par with or even surpassing supervised methods. They have also exhibited huge potential in cross-modal learning, such as the CLIP by [51]. However, we argue that these SSL methods are all built upon specific modalities with explicit topology and stationarity which URLOST goes beyond.

## 6 DISCUSSION

The success of most current state-of-the-art self-supervised representation learning methods relies on the assumption that the data has known stationarity and domain topology, such as the grid-like RGB images and time sequences. However, biological vision systems have evolved to deal with signals with less regularity so it can develop more efficient sensor arrangements. In this work, we explore unsupervised representation learning under a more general assumption, where the stationarity and topology of the data are unknown to the machine learning model and its designers. We argue that this is a general and realistic assumption for high-dimensional data in modalities of natural science. We propose a novel unsupervised representation learning method that works under this assumption and demonstrates our method's effectiveness and generality on a synthetic biological vision dataset and two datasets from natural science that have diverse modalities. We also perform a step-by-step ablation study to show the effectiveness of the novel components in our model.

During experiments, we found that density adjusted spectral clustering is crucial for the quality of representation learning. How to adjust the density and obtain a balanced clustering for any given data or even learning the clusters end-to-end with the representation via back-propagation is worth future investigation. Moreover, our current self-organizing layer is still simple though it shows effective performance. Extending it to a more sophisticated design and potentially incorporating it with various neural network architectures is also worth future exploration.

In summary, our method offers a handy and general unsupervised learning tool when dealing with high-dimensional data of arbitrary modality with unknown stationarity and topology, particularly

common in the field of natural sciences, where many present strong unsupervised learning baselines cannot directly adapt. We hope it can provide inspiration for work in related fields.

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

## A  APPENDIX

### A.1  MOTIVATION OF DENSITY ADJUSTED SPECTRAL CLUSTERING

Using the terminologies in high-dimensional statistics and functional analysis, the mutual information graph defined in section 2.2 corresponds to a compact Riemannian manifold $\mathcal{M}$ and the Laplacian matrix $L$ is a discrete analogous to the Laplace Beltrami operator $\mathcal{L}$ on $\mathcal{M}$. Minimizing the spectral embedding objective $tr(YLY^T)$ directly corresponds to the following optimization problem in function space:

$$\min_{||f||_{L^2(\mathcal{M})}} \int_{\mathcal{M}} ||\nabla f||^2 d\lambda \tag{5}$$

where $f(x) : \mathcal{M} \to [0,1]$ is the normalized signal defined on $\mathcal{M}$. We particularly want to write out the continuous form of spectral embedding so we can adapt it to non-stationary signal. To do so, we assume the measure $\lambda$ is absolutely continuous with respect to standard measure $\mu$. By apply the Radon-Nikodym derivative to equation 5, we get:

$$\int_{\mathcal{M}} ||\nabla f||^2 d\lambda = \int_{\mathcal{M}} ||\nabla f||^2 \frac{d\lambda}{d\mu} d\mu$$

where the quantitfy $\frac{d\lambda}{d\mu}$ is called the Radon-Nikodym, which is some form of denstiy function. Let $p(x) = \frac{d\lambda}{d\mu}$, we can rewrite the optimization problem as the following:

$$\min_{||f||_{L^2(\mathcal{M})}} \int_{\mathcal{M}} ||p(x)^{\frac{1}{2}} \nabla f(x)||^2 d\mu \tag{6}$$

The density function $p(x)$ on the manifold is analogous to the density adjustment matrix in equation 1. Standard approaches in equation 5 assume that nodes are uniformly distributed on the manifold, thereby treating $p(x)$ as a constant and excluding it from the optimization process. However, this assumption does not hold in our case involving non-stationary signals. Our work introduces a variable density function $p(x)$ for each signal, making it a pivotal component in building good representations for non-stationary signal. This component is referred to as *Density Adjusted Spectral Clustering*. Empirical evidence supporting this design is provided through visualization and ablation studies in the experimental section.

### A.2  SPECTRAL CLUSTERING ALGORITHM

Given a high dimensional dataset $S \in \mathbb{R}^{n \times m}$, Let $S_i$ be $i$th column of $S$, which represents the $i$th dimension of the signal. We create probability mass functions $P(S_i)$ and $P(S_j)$ and the joint distribution $P(S_i, S_j)$ for $S_i$ and $S_j$ using histogram. Let the number of bins be $K$. Then we measure the mutual information between $P(S_i)$ and $P(S_j)$ as:

$$I(S_i; S_j) = \sum_{l=1}^{K} \sum_{k=1}^{K} P(S_i, S_j)[l, k] \log_2 \left( \frac{P(S_i, S_j)[l, k]}{P(S_i)[l]P(S_j)[k])} \right)$$

Let $A_{ij} = I(S_i; S_j)$ be the affinity matrix, and let density adjustment matrix be $P$ defined in 2.2. Correlation is used instead of mutual information when dimension is really high, since compute mutual information is expensive.~~$p(i)$ be the density function defined in 4.~~ We follow the steps from [38] to perform spectral clustering with a modification to adjust the density:

Table 4: **Evaluation on foveated CIFAR-10 with varying hyperparameter for density function.** For each set of values of $\alpha$ and $\beta$, we perform density adjusted spectral clustering and run URLOST with the corresponding cluster. The evaluation of each trial is provided in the table.

|             | beta = 0  | beta = 2  |
| ----------- | --------- | --------- |
| alpha = 0   | 82.74 %   | 84.24 %   |
| alpha = 0.5 | 84.52 %   | 85.43 %   |
| alpha = 1.0 | 83.83 %   | 81.62 %   |

1. Define $D$ to be the diagonal matrix whose $(i,i)$-element is the sum of $A$'s $i$-th row, ~~$P$ be the diagnol matrix where $P_{ii} = p(i)$.~~ Construct the matrix $L = P^{\frac{1}{2}} D^{-\frac{1}{2}} A D^{-\frac{1}{2}} P^{\frac{1}{2}}$.

2. Find $x_1, x_2, \cdots, x_k$, the $k$ largest eigenvectors of $L$, and form the matrix $X = [x_1, x_2, \cdots, x_k] \in \mathbb{R}^{n \times k}$ by stacking the eigenvectors in columns.

3. Form the matrix Y from X by renormalizing each of $X$'s rows to have unit norms. (i.e. $Y_{ij} = X_{ij}/(\sum_i X_{ij}^2)^{\frac{1}{2}}$)

4. Treating each row of $Y$ as a point in $\mathbb{R}^k$, cluster them into $k$ clusters via K-means or other algorithms.

Some other interpretation of spectral embedding allows one to design a specific clustering algorithm in step 4. For example, [56] interprets the eigenvector problem in 6 as a relaxed continuous version of K-way normalized cuts problem, where they only allow $X$ to be binary, i.e. $X \in \{0,1\}^{N \times K}$. This is an NP-hard problem. Allowing $X$ to take on real value relaxed this problem but created a degeneracy solution. Given a solution $X^*$ and $Z = D^{-\frac{1}{2}} X^*$, for any orthonormal matrix $R$, $RZ$ is another solution to the optimization problem 6. Thus, [56] designed an algorithm to find the optimal orthonormal matrix $R$ that converts $X^*$ to discrete value in $\{0,1\}^{N \times K}$. From our experiment, [56] is more consistent than K-means and other clustering algorithms, so we stick to using it for our model.

### A.3  DATA SYNTHESIZE PROCESS

We followed the retina sampling approach described in [8] to achieve foveated imaging. Specifically, each retina ganglion cell is represented using a Gaussian kernel. The kernel is parameterized by its center, denoted as $\vec{x}_i$, and its scalar variance, $\sigma_i'^2$, i.e. $\mathcal{N}(\vec{x}_i, \sigma_i'^2 \mathbf{I})$, which is illustrated in Figure 5.A. The response of each cell, denoted as $G[i]$, is computed by the dot product between the pixel value and the corresponding discrete Gaussian kernel. This can be formulated as:

$$G[i] = \sum_n^N \sum_m^W K(\vec{x}_i, \sigma_i')[n, m] I[n, m]$$

where $N$ and $W$ are dimensions of the image, and $I$ represents the image pixels.

For foveated CIFAR-10, since the image is very low resolution, we first upsample it 3 times from $32 \times 32$ to $96 \times 96$, then use in total of 1038 Gaussian kernels to sample from the upsampled image. The location of each kernel is illustrated in Figure 5.B. The radius of the kernel scales proportionally to the eccentricity. Here, we use the distance from the kernel to the center to represent eccentricity. The relationship between the radius of the kernel and eccentricity is shown in Figure 5.C. As mentioned in the main paper, in the natural retina, retinal ganglion cell density decreases linearly with eccentricity, which makes the fovea much denser than the peripheral, unlike the simulated lattice we created. The size of the kernel should scale linearly with respect to eccentricity as well. However, for the low-resolution CIFAR-10 dataset, we reduce the simulated fovea's density to prevent redundant sampling. In this case, we pick the exponential scale for the relationship between the size of the kernel and eccentricity so the kernel visually covers the whole visual field. We also implemented a convolution version of the Gaussian sampling kernel to speed up data loading.

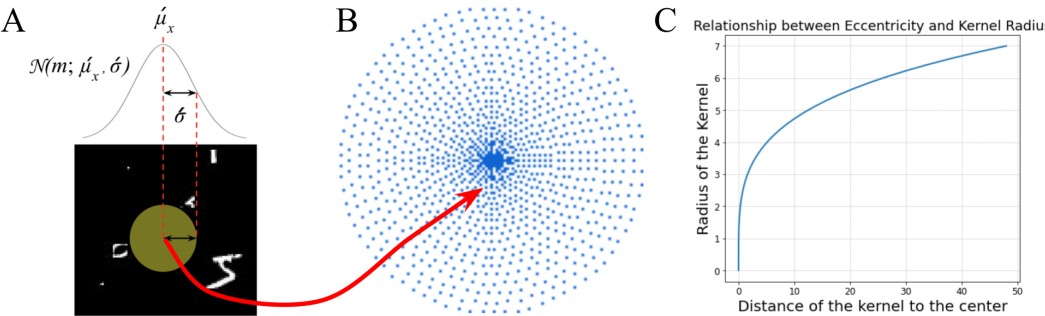

Figure 5: **Foveated retinal sampling** (A) Illustration of a Guassian kernel shown in [8]. Diagram of single kernel filter parameterized by a mean $\mu'$ and variance $\sigma'$. (B) the location of each Gaussian kernel is summarized as a point with 2D coordinate $\mu'$. In total, the locations of 1038 Gaussian kernels are plotted. (C) The relationship between eccentricity (distance of the kernel to the center) and radius of the kernel is shown.

### A.4    DENSITY ADJUSTED SPECTRAL CLUSTERING ON FOVEATED CIFAR10 DATASET

We provide further intuition and visualization on why density adjusted spectral clustering allows the model to learn a better representation on the foveated CIFAR-10 dataset.

As shown in Figure 5, the kernel at the center is much smaller in size than the kernel in the peripheral. This makes the kernel at the center more accurate but smaller, which means it summarizes less information. Spectral clustering with constant density will make each cluster have a similar number of elements in them. Since the kernel in the center is smaller, the cluster in the center will be visually smaller, than the cluster in the peripheral. The effect is shown in Figure 6. Moreover, since we're upsampling an already low-resolution image (CIFAR-10 image), even though the kernel at the center is more accurate, we're not getting more information. There, to make sure each cluster has similar information, the clusters in the center need to have more elements than the clusters in the peripheral. In order to make the clusters at the center have more elements, we need to weight the clusters in the center more with the density function. Since the sampling kernels at the center have small eccentricity and are more correlated to their neighbor, increasing $\alpha$ and $\beta$ will make sampling kernels at the center have higher density, which makes the cluster at the center larger. This is why URLOST with density adjusted spectral clustering performs better than URLOST with constant density spectral clustering, which is shown in Table 4. Meanwhile, setting $\alpha$ and $\beta$ too large will also hurt the model's performance because it creates clusters that are too unbalanced.

### A.5    SELF-ORGANIZING LAYER LEARNS INVERSE PERMUTATION

For *locally-permuted CIFAR-10*, we divide each image into patches and locally permute all the patches. The $i$-th image patch is denoted by $x^{(i)}$, and its permuted version, permuted by the permutation matrix $E^{(i)}$, is expressed as $E^{(i)}x^{(i)}$. We use linear layers to parameterize the self-organizing layers. Let $g(x, W^{(i)}) = W^{(i)}x$ denotes the $i$th element of the self-organizing layer. We're providing the proof for the statement related to the visual evidence shown in Section 4.1

Statement: If the self-organizing layer effectively aligns the input sequence, then $E^{(i)T}w^{(i)}$ should exhibit visual similarities.

Proof: we first need to formally define what it means for the self-organizing layer to effectively align the input sequence. Let $\mathbf{e}_k$ denote the $k$th natural basis (one-hot vector at position $k$), which represents the pixel basis at location $k$. Permutation matrix $E^{(i)}$ will send $k$th pixel to some location accordingly. Mathematically, if the projection layer effectively aligns the input sequence, it means $g(E^{(j)}e_k, W^{(j)}) = g(E^{(i)}e_k, W^{(i)})$ for all $i, j, k$. We can further expand this property to get the

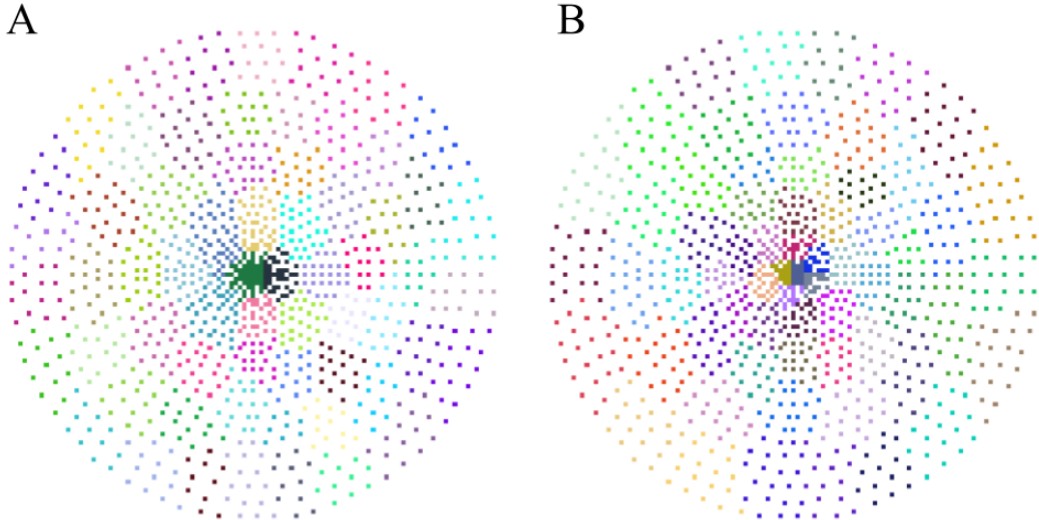

Figure 6: **Effect of density adjusted clustering.** Eccentricity-based sampling lattice. The center of the sampling lattice has more pixels which means higher resolution compared to the peripheral. (A) Result of density adjusted spectral clustering ($\alpha = 0.5, \beta = 2$). Clusters in the center have more elements than clusters in the peripheral. But clusters look more visually similar in size than B. (B) Result of uniform density spectral clustering ($\alpha = 0, \beta = 0$). Each cluster has a similar number of elements in them but the clusters in the center are much smaller than the clusters in the periphery.

following two equations:

$$g(E^{(i)}e_k, W^{(i)}) = W^{(i)}E^{(i)}e_k$$
$$g(E^{(j)}e_k, W^{(j)}) = W^{(j)}E^{(j)}e_k$$

for all $i, j, k$. Since the above equation holds for all $e_k$, by linearity and the property of permutation matrix, we have:

$$W^{(i)}E^{(i)} = W^{(j)}E^{(j)}$$
$$E^{(i)T}W^{(i)} = E^{(j)T}W^{(j)}$$

This implies $E^{(i)T}w^{(i)}$ should exhibit visual similarities for all $i$.

### A.6 TRAINING AND EVALUATION DETAILS

$\beta$-**VAE**. $\beta$-VAE was trained for 1000 epochs and 300 epochs on the V1 neura l and TCGA gene expression respectively. We use the Adam optimizer with a learning rate of $0.001$ and a cosine annealing learning rate scheduler. The encoder is composed of a 2-layer MLP with batch normalization and LeakyReLU activation. Then two linear layers are applied to get the mean and standard for reparameterization. The decoder also has a 2-layer MLP, symmetric to the encoder but using standard ReLU activation and no batch normalization. We tried out different hyperparameters and empirically found this setting gives the best performance.

**MAE**. MAE follows the official implementation from the original paper. For CIFAR10, we ran our model for 10,000 epochs. We use Adam optimizer with learning rate 0.00015 and a cosine annealing. To fit in our tasks, we use 8 layers encoder and 4 layer decoder with hidden dimension 192. The ViT backbone can take different patch sizes are we indicated them accordingly in Table 1. ViT(Pixel) means treating each pixel as a patch, so essentially the patch size is 1. This is also used for the real-world high-dimensional dataset since no concept of patch is defined in the signal space.

For V1 neural recording and TCGA gene expression task, we use 4 layers encoder and 2 layer. We use hidden dimension 1380 for 1000 epochs and hidden dimension 384 with 3000 epochs for V1 neural recording and TCGA dataset task. The hidden dimension and the number of epochs we used for MAE is greater than $\beta$-VAE. However, when we use the same parameters on $\beta$-VAE, we did not seem to find a performance gain. Training transformer usually require large number of data. For example, the original transformer on vision is pretrained over 14M images.

**URLOST MAE** The parameter of URLOST MAE is the same as MAE except for the specific hyper-parameter in the method section. For CIFAR10, we use $K = 20$, $\alpha = 0.5$ and $\beta = 2$. We set number of clusters to be 64. For V1 neural recording, we use $K = 15$, $\alpha = 0$ and $\beta = 1$. We set number of clusters to be 200. For TCGA dataset, we use $K = 10$, $\alpha = 0$ and $\beta = 1$. We set number of clusters to be 32.

## A.7 VISUALIZING THE WEIGHT OF SELF-ORGANIZING

As explained in the previous section (Appendix A.5) and visualized in Figure 7, we can visualize the weights of the learned self-organizing layer when trained on the locally-permuted CIFAR-10 dataset. If we apply the corresponding inverse permutation $E^{(i)T}$ to its learned filter $W^{(i)}$ at position $i$, the pattern should show similarity across all position $i$. This is because the model is trying to align all the input clusters. We have shown this is the case when the model converges to a good representation. On the other hand, what if we visualize the weight $E^{(i)T}W^{(i)}$ as training goes on? If the model learns to align the clusters as it is trained for the mask prediction task, $E^{(i)T}W^{(i)}$ should become more and more consistent as training goes on. We show this visualization in Figure 7, which confirms our hypothesis. As training goes on, the pattern $E^{(i)T}W^{(i)}$ becomes more and more visually similar, which implies the model learns to gradually learn to align the input clusters.

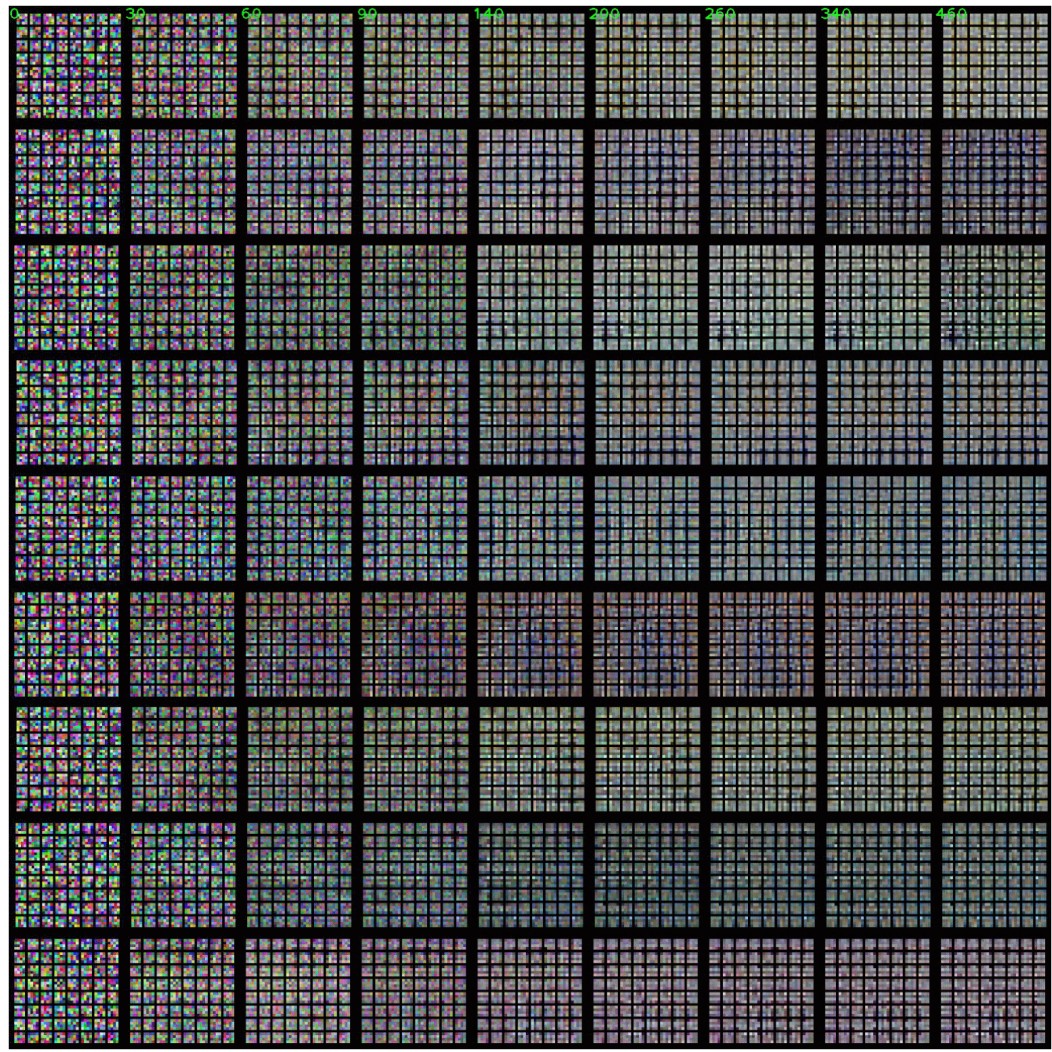

Figure 7: **Visualize the weight of the self-organizing layer after applying inverse permutation.** A snapshot of $E^{(i)T}W^{(i)}$ is shown at different training epoch. The number of epochs is shown on the top row. Each figure shows one column of the weight of the self-organizing layer, at different positions, i.e. $W^{(1)}_{:,k}$, where $k$ is the column number and $i$ is the position index. In total, 9 columns are shown.

