# OpenReview forum: "URLOST: Unsupervised Representation Learning without Stationarity or Topology"
_ICLR.cc/2024/Conference — Submitted to ICLR 2024_

### Official Review · Reviewer_qyGv · 2023-10-28

**Soundness:** 2 fair
**Presentation:** 4 excellent
**Contribution:** 2 fair
**Rating:** 6
**Confidence:** 3

**Summary:**

The paper introduces a framework for unsupervised representation learning that overcomes the limitations of stationarity and topology. It combines a learnable self-organizing layer, density adjusted spectral clustering, and masked autoencoders to learn representations from high-dimensional data without relying on stationarity or topology. The model is evaluated on various datasets, including simulated biological vision data, neural recordings from the primary visual cortex, and gene expression datasets. Results show that the proposed method works well. The approach demonstrates promise for building unsupervised representations across high-dimensional data modalities, including modalities in natural sciences like chemistry, biology, and neuroscience.

**Strengths:**

- The paper is well-written. A clear structure can enhance the readability and understanding of a paper. However, the decision to reuse a large Figure 1 from multiple papers raises questions about the originality and visual presentation of the work. Creating a unique figure would enhance the paper's credibility and make it more visually cohesive.

- The underlying idea is recognized as interesting and well-motivated. The paper successfully conveys the rationale behind each stage of the proposed method.

- The paper conducts various experiments on different datasets, including simulated biological vision data, neural recordings, and gene expression datasets. This demonstrates the versatility and applicability of the proposed method across different modalities.

**Weaknesses:**

- The combination of well-known techniques in a pipeline is noted as a potential limitation. The lack of an end-to-end deep learning architecture is highlighted, questioning the level of innovation in the proposed method.

- The absence of information on seed-related results makes it challenging to fully interpret and reproduce the reported results. More comprehensive reporting of experimental methodologies would strengthen the paper's scientific rigor. More transparency in the experimental methodology and inclusion of uncertainty statistics would enhance the robustness of the reported results.

- The real-world dataset results are currently unconvincing. The reported accuracy metrics are comparable, and the absence of uncertainty statistics diminishes the robustness of the claims. More detailed analysis and statistical measures are needed to support the performance claims on real-world datasets.

**Questions:**

Refer to the comments above.

---

> ### Author Response · Authors · 2023-11-23
> **Reply to Reviewer qyGv (1/2)**
>
> __Q1__: The combination of well-known techniques in a pipeline is noted as a potential limitation. The lack of an end-to-end deep learning architecture is highlighted, questioning the level of innovation in the proposed method.
>
> __A1__:
> Thanks for pointing this out. We acknowledge that our method consists of a combination of existing techniques. We would like to note that not all innovative methods are end-to-end deep learning. Many innovations happen in a way that combines existing methods, e.g. spectral clustering. Especially when working on novel problems, effective combinations that provide straightforward methods have their novel contributions and can be beneficial to the research community.
>
> Specifically, our work is dealing with a novel problem. Unsupervised representation learning without knowing the explicit topology or stationarity of the high-dimensional data is largely overlooked by the community. As for the method, we think it is intuitive and the combination is natural. To work with arbitrary high-dimensional signals, we first group the dimensions that share high mutual information (clustering), then align the signals defined on these clusters into the same feature space such that the inner product reflects their similarity (self-organization). Then the signals can work like image patches for MAE self-supervised learning. Overall, we provide a straightforward method to this novel problem, and based on the intuition, the combination like this is rather powerful.
>
> In terms of the specific techniques used, we want to note that it is not a simple A+B+C combination of existing techniques. During our exploration we found simple combinations of spectral clustering, patch embedding and MAE would not work, instead, we propose to adjust the density of the Laplacian matrix during clustering, and use non-shared projection layer to replace the uniform patch embedding. These two techniques together allow our method to estimate the topology and stationarity of the input signal, and align the signals defined on the clusters to a latent space where the inner product can reflect their similarity, just like the image patches. In this way we can leverage MAE to do self-supervised learning. In section 4, we explained the intuition and ablation studies that support this design. We have also revised the method section to better explain our design choices (marked in blue).
>
> Overall our work provides a straightforward method to tackle an important novel problem. It has significant meaning to have a method that can do unsupervised representation learning on high-dimensional signals without knowing its topology and stationarity. It will enable the power of self-supervised learning on arbitrary high-dimensional data in diverse data sources and modalities, like we saw in language and images in recent years. So it is a novel problem that deserves further exploration. We do not claim that our current method has fully solved this problem, but it rather serves the purpose of proposing this new problem, taking a first step to solve this new problem and can be a seed to simulate future research. We hope our work can inspire more people to tackle this problem, and this will be our future direction as well.
> In that regard, we think our method has its merit and novel contribution.
>
> ---
>
> __Q2__: The absence of information on seed-related results makes it challenging to fully interpret and reproduce the reported results. More comprehensive [...] More transparency in the experimental methodology.
>
> __A2__: Thanks for pointing this out! We provide new experiment results where we vary the seed numbers and try 15 seeds and report the performance mean and variance w.r.t seed choices. We have also revised the paper to add the new results with confidence intervals for our method and the strongest baseline. Due to the time and computation constraints, we will provide more seed-related results on all the baseline methods for the camera-ready version.
>
> We appreciate your question on reproducibility and transparency. We share the same value on their importance. In the submission, we omit details on hyperparameters used to train and evaluate the model for a more clean and concise presentation. In the revised manuscript, we have added many of these details in the Appendix including the hyperparameters and implementation details. We have also listed the seeds we used for evaluation. All revisions are marked in blue. We believe the paper is reproducible with this additional information. We will also release a repo with the complete source code for the complete reproduction of this work after the paper is accepted. We will attach a link to the codebase in the camera-ready version. Please let us know if you have any other suggestions on helping reproducibility, so we can revise for the camera-ready version of the paper.

---

> ### Author Response · Authors · 2023-11-23
> **Reply to Reviewer qyGv (2/2)**
>
> __Q3__: The real-world dataset results are currently unconvincing. The reported accuracy metrics are comparable, and the absence of uncertainty statistics diminishes the robustness of the claims. More detailed analysis and statistical measures are needed to support the performance claims on real-world datasets.
>
> __A3__:
> Thanks for the suggestion! We have added new results including 95% confidence interval on the real-world dataset to the revised manuscript obtained by running 15 experiments by using different seed numbers. The results are also attached here:
>
> | Method |    V1 Response Decoding Acc  |   TCGA Classification Acc  |
> | :-----------:| :-----------: | :-----------: |
> |  $\beta$-VAE | 75.64 \% $\pm$ 0.11 \% |   94.15\% $\pm$ 0.24\%  |
> | URLOST MAE | 78.75\% $\pm$ 0.18 \% | 94.90\% $\pm$ 0.25 \% |
>
> Since the 95% confidence intervals of URLOST on both tasks uniformly outperform the interval of $\beta$-VAE, we think URLOST gets reliably better performance on these datasets. Due to the time and computation constraint, we could only report these statistics for our method and the $\beta$-VAE because it is the strongest baseline with closest performance. But for completeness, we will provided the same statistics for all the baseline methods in the camera-ready version. We would be happy to continue to test the method on new dataset and tasks and this is our future direction as well.
>
>
> ---
>
> __Q4__: the decision to reuse a large Figure 1 from multiple papers raises questions about the originality and visual presentation of the work. Creating a unique figure would enhance the paper's credibility and make it more visually cohesive.
>
> __A4__: Thank you for the suggestion regarding Figure 1. Figure 1 was used for an illustrative purpose in our first submission. It is used to introduce the new problem of unsupervised learning without topology and stationarity and to do so, we use figure 1 to make comparison with the the schemes of existing SSL techniques and also connect it to the inspirations from neuro perception. That is why you're seeing many familiar structures. However, none of the parts in figure 1 was directly copied and pasted from previous papers. Only the pictures of specific data points or dataset are borrowed and to which we add their references. We decided to use these real pictures so that the illustration is less abstract to the readers. Overall the figure is uniquely designed and drawn on our own. For our own method and visualization, we have presented figure 2, 3, 4 in the main paper and 6, 7 in the appendix. Drawing good and informative figures for an academic paper is always a creative yet challenging process and we are always happy to hear any suggestions and continue to improve it.

---

> > ### Comment · Reviewer_qyGv · 2023-11-23
> >
> > I appreciate your consideration of my concerns in this revision. The authors have addressed the majority of those issues, leading me to adjust my score to reflect the positive changes made.

---

### Official Review · Reviewer_JGWe · 2023-10-31

**Soundness:** 2 fair
**Presentation:** 2 fair
**Contribution:** 2 fair
**Rating:** 5
**Confidence:** 1

**Summary:**

The paper introduces a novel method for unsupervised representation learning which can handle signals lacking an explicit topology. The method is then evaluated on synthetic biological vision datasets,

**Strengths:**

The problem considered by the author seems relevant and their method intriguing, but I do not have enough experience in the field to give a proper judgement of the strengths.

**Weaknesses:**

See above. Additionally, some concepts introduced in the text are poorly defined and impossible to understand for a non-expert.

1. In section 2.2, what are the 'dimensions' i and j? What is the manifold M? How do you define a Laplacian based on A?
2. After equation 1, what density does p(x) represent?
3. Section 2.3, the intended meaning of alignment is not clear to me. The idea of directly solving the alignment problem with low-level statistics is completely obscure: What is the alignment problem?
4. What are the metrics used for evaluation in Table 1?

**Questions:**

see above.

---

> ### Author Response · Authors · 2023-11-22
> **Reply to Reviewer JGWe (1/3)**
>
> We apologize for the confusion. We left some of these technical details in the appendix to fit more content in the main draft. However, we agree with the reviewer that this makes the method section confusing. As suggested by the reviewer, we added in more detail in the method section and result section, including better notation, clearer explanations and motivations. These changes are intended to enhance the paper's readability, not just for experts in the field but for a broader audience as well. We are also open to any further suggestions you might have to improve the paper's clarity and accessibility.
>
> We made a major revision on section 2.2. We move the functional analysis motivation of density adjusted spectral clustering to the appendix. Instead, we give a clear definition of the dataset and the exact formulation of density-adjusted spectral embedding in section 2.2. We also move the density function to section 2.2. We hope this makes the narrative cleaner and less confusing. In section 2.3, we add an explanation for the “alignment problem”. In the experimental section, we add a brief description of the evaluation procedure in the table.

---

> ### Author Response · Authors · 2023-11-22
> **Reply to Reviewer JGWe (2/3)**
>
> Here’s the reply to each of author’s questions:
>
> __Q1__: In section 2.2, what are the 'dimensions' i and j? And How do you define a Laplacian based on A?
>
> __A1__: Given a high dimensional dataset $S \in \mathbb{R}^{n \times m}$, where n is the number of samples and m is the dimension of the data. let $S_i$ be $i$th column of $S$. i, j represents the ith dimension and jth dimension of the signal. We can estimate the mutual information between the dimension and jth dimension of the signal by using samples in $S_i$ and $S_j$ as I(S_i;S_j).
>
> Given the affinity matrix A, we define the Laplacian operator as $L = D - A$.
>
> ---
>
> __Q2__:  What is the manifold M?
>
> __A2__: The manifold $\mathcal{M}$ here implies the lower-dimensional subspace where the high-dimensional signal lies. For example, 2-d image with thousands of pixel can be considered as a high dimensional signal, but lies on a 2-d manifold.
>
> In the spectral embedding, the Laplacian matrix L describes a graph that lies on the manifold. The Laplacian matrix $L$ is a discrete analogous to the Laplace Beltrami operator $\mathcal{L}$ on a compact Riemannian manifold $\mathcal{M}$. $\mathcal{M}$ is the hypothetical continuous domain of the signal induced by the graph. Minimizing $ \int_M ||\nabla f||^2 d\lambda $ corresponds to solving the continuous version of spectral embedding.
>
> tldr: The only reason we want to write the continuous version of spectral embedding is so that it explains the motivation of our algorithm, namely density adjusted spectral clustering. To avoid confusion, we talk about the precise spectral embedding algorithm we solved in section 2.2 (the discrete version). The “manifold” interpretation is now moved to the appendix for cleaner narrative.
>
> ---
>
> __Q3__: After equation 1, what density does p(x) represent?
>
> __A3__:Thanks for pointing this out! We acknowledge that the meaning of density function p(x) is only introduced in the latter section in section 4.2. We move the definition of density function in much earlier section (2.1) to avoid confusion.
>
> In the revised version, we use the notation p(i) instead of p(x). i is used for indexing the “ith” node. Concretely, p(i) will define a diagonal matrix $P = diag(p(i))$, which is used to reweigh the Laplacian matrix L during spectral clustering. This is similar to the normalized laplacian matrix.
>
> p(i) defines the “density” of ith node. We define it as $p(i) = q(i)^{\alpha} n(i)^{-\beta}$, where $n(i)$ is defined as the sum of top k mutual information between node $i$ and other nodes. $q(i)$ is the prior density which depends on the dataset.
>
> The intuition is that we want to set p(i), such as when we apply this P to the Laplacian matrix L, the density of each node is roughly the same to achieve stationarity. For example, if node i originally has high density, then it will give higher mutual information to its neighbors. Then we want to make p(i) smaller, which is reflected by the definition of p(i). Some dataset have prior knowledge on the density. For example, in foveated CIFAR10, the density of each node should scale with eccentricity (retinal ganglion cells are denser in the fovea and sparser in the peripheral). We define $q(i)$ as the distance from the sampling kernel at $i$th node to the center of the sampling lattice. For the V1 neuron recording and TCGA dataset, since we have no prior knowledge on the nodes in this dataset, we set $\alpha = 0$.
>
> **unfortunately, some equation cannot be rendered in openreview. please check in section 2.2 for the details.

---

> ### Author Response · Authors · 2023-11-22
> **Reply to Reviewer JGWe (3/3)**
>
> __Q4__: Section 2.3, the intended meaning of alignment is not clear to me. The idea of directly solving the alignment problem with low-level statistics is completely obscure: What is the alignment problem?
>
> __A4__:Thanks for pointing this out! We acknowledge that the definition of this alignment was not provided clearly, and we have revised Section 2.3 accordingly, marked in blue. In short, the goal of this alignment is to transform the signals defined on different clusters into a space such that their inner products reflect their similarity. First, let's take a look at an intuitive example: Given an image, we divide it into a set of image patches of the same size and apply different permutations to these image patches. Then, their inner product will no longer reflect their similarity properly. Similarly, the spectral clustering introduced in Section 2.2 was used to group pixels into clusters. The image signal defined on the clusters becomes our “patches.” However, since the ordering of pixels in each cluster is arbitrary, if we take two patches defined on two different clusters of pixels, their inner product is also arbitrary due to the ordering mismatch. The self-organization layer introduced in Section 2.3 is to linearly project each “patch” into an aligned space such that their inner products reflect their similarity again. We introduce this additional layer and leverage the self-attention mechanism of the Transformer to achieve this goal. In Transformers, since self-attention depends on the inner products between different “patches,” the self-organization layer tends to learn the correct linear projections to align the “patches” into a space, where their inner products reflect their similarity. In Section 4.1 and particularly Figure 4, we provide a visualization for this process. In Appendix A.6 and Figure 7, we provide additional analysis and visualization.
>
>
>
> ---
>
> __Q5__: What are the metrics used for evaluation in Table 1?
>
> __A5__:  The evaluation metric we used is linear probing since it is the default metric for evaluating the representation quality of the recent self-supervised learning models [1] [2] [3]. We first train our unsupervised learning model on the data. This is called the pretraining step. Then we encode each data point to an embedding vector with our pretrained encoder. Finally, we perform logistic regression on embedding vectors while keeping the encoder frozen from being updated with their labels and recording the accuracy. We add the concise description of our evaluation procedure in Table 1 as suggested by the reviewer, to make the table more self-contained.
>
> [1] Chen, Ting, et al. "A simple framework for contrastive learning of visual representations." International conference on machine learning. PMLR, 2020.
>
> [2] He, Kaiming, et al. "Masked autoencoders are scalable vision learners." Proceedings of the IEEE/CVF conference on computer vision and pattern recognition. 2022.
>
> [3] He, Kaiming, et al. "Momentum contrast for unsupervised visual representation learning." Proceedings of the IEEE/CVF conference on computer vision and pattern recognition. 2020.

---

### Official Review · Reviewer_EzDe · 2023-11-05

**Soundness:** 3 good
**Presentation:** 3 good
**Contribution:** 3 good
**Rating:** 6
**Confidence:** 4

**Summary:**

The authors propose to extend the utility of unsupervised representation learning under the relaxation of conventional assumptions related to stationarity and topology. They are inspired by biological visualization systems and assert that the existing ansatz are not sufficiently inclusive of general high-dimensional data.

**Strengths:**

The authors ground their approach in a concrete example that is returned to throughout thereby helping the reader to build a stronger understanding and intuition for the work.

Method is demonstrated for multiple modalities suggesting its broader utility. The method seems to possess impactful new capabilities.

Figures and result tables are clear with reasonably informative captions-- something that is not always the case and greatly appreciated.

**Weaknesses:**

The core findings and impact to the field are not clearly identified in the introduction. They include the questions they aim to answer and a high-level description of their approach but concrete claims or impact are omitted. While the reviewer was very intrigued by the paper, the primary weakness was the lack of a clear statement of the "so what?"

While helpful for building intuition, the classification accuracy studies are less compelling as they feel more contrived yet they get the bulk of the attention. From this viewer's perspective, the presented method's potential to be able to infer the permutation or inform about the topology or stationarity of data is the more novel component. Although it may just be a personal preference, I think highlighting and demonstrating those unique abilities would emphasize the novelty and utility of the method.

Results tables should include confidence intervals.

In this reviewer's estimation there is insufficient information available for reproducibility within the main body of the paper.

**Questions:**

What are the top three claims or novel contributions of the work from the authors' perspective?

What can the proposed approach tell you about the stationarity and topology of the data it is provided?

Are there connections between this work and encryption/decryption?

---

> ### Author Response · Authors · 2023-11-22
> **Reply to Reviewer EzDe (1/4)**
>
> __Q1__: “.. concrete claims or impact are omitted .. What are the top three claims or novel contributions of the work from the authors' perspective?”
>
> __A1__:
>
> This is an important comment. We add our major contributions in the last paragraph of the introduction, which contains core findings and impact. We rephrase our contributions as core findings and impact as the following:
>
> 1. We identify an important self-supervised learning problem that is largely ignored by the machine learning community: how to build unsupervised representation for general high-dimension data? High-dimensional data is prevalent in everyday life, scientific research, and nature.
> 2. And, we propose a straightforward yet effective method that provides a step for solving this problem, which combines intuition for high dimensional statistics, bio-inspired design and state-or-the-art self-supervised learning method. Specifically, our model is inspired by formation of retinotopy and how visual systems compute with retinotopic input.
> 3. We show our model can deal with diverse modalities and have numerous applications. For example, it can serve as a vision foundation model with more efficient sensor arrangement. It can also be used as a hypothesis testing tool on neural data and a cancer diagnosis tool.

---

> ### Author Response · Authors · 2023-11-22
> **Reply to Reviewer EzDe (2/4)**
>
> __Q2__: “Method’s potential to inform about the topology or stationarity of data is the more novel component.” “What can the proposed approach tell you about the stationarity and topology of the data it is provided?”
>
> __A2__: This is a unique perspective, and we highly appreciate this perspective. Method’s potential to explicitly inform about the topology or stationarity of data is not emphasized in this paper. This is because our goal is to develop a high level self-supervised learning algorithm for high dimensional data, where the model learns a good representation of the data for perceptual level downstream tasks. Reviewers suggest one of the potential impacts of work is that the model we developed can be used as an unsupervised learning method to recover topology and enforce stationary on the signal. We found this perspective very interesting so we emphasized this perspective in the paper.
>
> The proposed approach tell us the following thing about the stationarity and topology of the data:
>
> First, we show we can infer the coarse topology from spectral clustering as shown in Figure 3. Spectral clustering provides dimension grouping, i.e. which group of dimensions should be processed together. However, this is a coarse topology because we did not define the relationship between two dimensions if they lie in different groups. In other words, if we shuffle the dimension in two groups, the coarse topology we recovered still remains the same.
>
> We then show in Figure 4 that we can recover the fine-grained topology by optimizing the self-organizing layer. In this example, a self-organizing layer learns how to permute elements in each group. This way, each dimension in a group is aligned with its corresponding dimension in the other group. In other words, given two groups, the ith dimension of the first group and the ith dimension of the second group now correspond to the same spatial position in an image patch. The local topology is recovered because each element in a group now has a meaningful coordinate. However, we cannot directly visualize the effect of a self-organizing layer. This is because it does not only help me un-permute the group, it also projects the image into a latent space. The latent space cannot be easily visualized. This is why we choose to visualize each filter learnt by the self-organizing layer in section 4.1.
>
> As for stationarity, it’s hard to make a precise and mathematical definition on what statistic  should be stationarity. However, we did show an intuitive example. In appendix 4, figure 6, we show that if we use out of the box spectral clustering on foveated and permuted images, the cluster we get at the center of the image is very small and the cluster we get at the peripheral is very large. Actually, the number of pixels in the center cluster is the same as the number of pixels in the peripheral. The difference is that the center pixel is smaller and the peripheral pixel is larger, which causes the visual size difference among these clusters. Here, we can roughly define a statistic for each cluster as “the area this cluster sampled from the image.” We want this statistic to be stationary. If we adjust the density for the spectral clustering, the area sampled by the center cluster is much smaller than the cluster in the peripheral, which makes the statistic non-stationary. The density adjustment makes the area of these clusters more uniform, which makes the statistic more stationary.
>
> Finally, we want to emphasize that we did not ask the self-organizing layer to recover topology or pixel to pixel alignment. It is being optimized for the “mask-token-prediction” task. However, it does choose to learn to recover topology which is very interesting.

---

> ### Author Response · Authors · 2023-11-22
> **Reply to Reviewer EzDe (3/4)**
>
> __Q3__:  “classification accuracy studies .. feel more contrived yet they get the bulk of the attention”
>
> __A3__:
>
> Current experiment on using our pretrained model to do classification shows our model learns a good representation of high dimensional data. We agree with the reviewer that this way of building representation is “black-box” and less interpretable. This has been a long-existing problem of neural network based unsupervised learning. However, this perspective of building good representations for downstream tasks is also highly required and appreciated by the machine learning community in general.
>
> We want to emphasize combining spectral clustering, self-organizing layer and masked autoencoder is a novel component of our work. This combination is simple, natural yet effective. There are other works on recovering the topology of the signal, like  [1] or [2]. However, these works cannot be effectively integrated with state-of-the-art self-supervised learning algorithms. For example,  [1] uses manifold learning to infer the 2-D position of each pixel. People try to feed the “recovered topology” to a graph neural network (GNN), but suffer from inherent scalability issues on using GNN to do unsupervised learning. We add this perspective in our related work section.
>
> [1] Roux, N, et al. 2007. Learning the 2-D topology of images.
>
> [2] Kohonen, T. 1990. The self-organizing map
>
> ---
>
> __Q4__: Could the author provide confidence intervals for the table?
>
> __A4__:
>
> Sure. This is a very good suggestion. Many aspects of the algorithm are stochastic, for example, the spectral clustering solver, SGD based optimizer for model training and linear classifier training, train/test set splitting. Repeating the training and evaluation on computer vision dataset is computationally expensive and usually not provided in other work. Moreover, since all our evaluation results on CIFAR10 already seem significant, we will provide the confidence interval for neural response decoding dataset and gene expression dataset.
> In these two dataset, out of the three baselines we provided,  beta-VAE and our URLOST are the strongest and outperform other baseline significantly. Thus, we first provide the confidence interval for URLOST and beta-VAE. For the camera-ready version paper, we will provide confidence intervals for all baseline methods.
>
> We generate the confidence interval in the following way:
> We pick 15 seeds randomly, and for each run, we plug the same seed into all seed-related functions in our code, including eigensolver, discretize algorithm for spectral clustering, data split, numpy seed, pytorch seed, and logistic regression function for linear probing evaluation.
> We pretrain the model from scratch and record the evaluation accuracy of each run.
> We then calculate the 95% confidence interval from the recorded accuracies. Results are attached below and also updated in the table in our revised manuscript:
>
>
> | Method |    V1 Response Decoding Acc  |   TCGA Classification Acc  |
> | :-----------:| :-----------: | :-----------: |
> |  $\beta$-VAE | 75.64 \% $\pm$ 0.11 \% |   94.15\% $\pm$ 0.24\%  |
> | URLOST MAE | 78.75\% $\pm$ 0.18 \% | 94.90\% $\pm$ 0.25 \% |
>
>
> The results with confidence intervals show that URLOST reliably outperforms beta-VAE since their 95% confidence intervals do not overlap for both of the tasks.
>
> ---
>
> __Q5__: There is insufficient information available for reproducibility within the main body of the paper.
>
> __A5__: We agree with this comment. We omit details on hyperparameters used to train and evaluate the model for a more clean and concise presentation. We added many of these details in the Appendix A6 including hyperparameters and implementation details. We have also listed the seeds we used for evaluation. We will provide a repo containing all our code before the camera-ready if the paper is accepted. We believe the paper is reproducible with this additional information. Please let us know if you have any other suggestions on helping reproducibility, so we can revise for the camera-ready version of the paper.

---

> ### Author Response · Authors · 2023-11-22
> **Reply to Reviewer EzDe (4/4)**
>
> __Q6__: Are there connections between this work and encryption/decryption?
>
> __A6__: This is a very interesting perspective! We did not make this connection during the process of this work. But conceptually the idea of rediscovering topology and stationarity from the data where the information is covered, and further reconstructing the data, can have a logical connection to the encryption/decryption. For now, we are not sure whether this can be used readily as a way for encryption/decryption since the “key” will not be secured in that it can be obtained by learning from data from the same distribution as the data of your encrypted message. For example, if some images are shuffled up for an encrypted transmission, anyone with access to a large image set could learn a decryption model, not exclusively the designated receiver. Unfortunately, none of the authors is an expert in encryption/decryption so this is only a preliminary answer. However, we really appreciate this question and the valuable idea proposed by the reviewer, and we would love to further discuss and explore new impact and potential application of our method.

---

### Author Response · Authors · 2023-11-22
**General reply for reviewers**

We thank our reviewers for their encouraging comments, helpful suggestions, and insightful questions. All reviewers think our method is intriguing. Both R1 and R3 recommends accepting our paper based on the strong motivation and impactful capabilities it entails. R1 EzDe says “The authors ground their approach in a concrete example”, “.. demonstrated for multiple modalities suggesting its broader utility.” “.. possess impactful new capabilities” R2 JGWe says “The problem seems relevant and their method intriguing.” R3 qyGv says “The paper is well-written,” “.. idea is recognized as interesting and well-motivated,” and “.. demonstrates the versatility and applicability of the proposed method across different modalities.”

Reviewers’ questions and suggestions are very constructive. They help us revise the paper to deliver key messages better. Both R1 and R3 suggest we should highlight the core findings and impact of this work. We modified our introduction according to their suggestions and added a paragraph talking about the contributions of this work. Reviewers also suggest we should improve reproducibility and add uncertainty statistics to strengthen the robustness of our claim.  We have added experiments with various seed numbers and updated the experiment table with confidence intervals. The new results are also attached to corresponding questions in the individual responses. We have also provided more implementation details in the revised manuscript. We provide an overview of the revision alongside a revised paper. And we will release our source code to improve reproducibility. The added parts are marked as blue in the revision, whereas strike lines mark the removed parts, and minor grammar correction is not recorded.

Main Contribution of this work:

1. We identify an important self-supervised learning problem that is largely ignored by the machine learning community: how to build unsupervised representation for general high-dimension data? High-dimensional data is prevalent in everyday life, scientific research, and nature.
2. And, we propose a straightforward yet effective method that provides a step for solving this problem, which combines intuition for high dimensional statistics, bio-inspired design and state-or-the-art self-supervised learning method. Specifically, our model is inspired by formation of retinotopy and how visual systems compute with retinotopic input.
3. We show our model can deal with diverse modalities and have numerous applications. For example, it can serve as a vision foundation model with more efficient sensor arrangement. It can also be used as a hypothesis testing tool on neural data and a cancer diagnosis tool.

Here we provide an overview of the changes made in the revision:

1. Section 1: A concise summary of the contributions is added to the introduction.
2. Section 1: Add more motivation/impact from computational neuroscience’s perspective, improve the discussion more clearly about
3. contribution and impact in the introduction.
4. Section 2: We revise Section 2.2 for better introduce density adjustment for spectral clustering. We move the functional analysis motivation to the appendix to reduce confusion and improve readability.
5. Section 2:  We provide an explanation for the ``alignment problem’’ in Section 2.3, to reduce confusion and improve readability.
6. Section 3: For vision tasks in Section 3.1, we added more experiment details and discussion on the density function in the spectral clustering, to improve the transparency and reproducibility.
7. Section 3: We modified the caption of Table 1 to clarify the evaluation metric of the experiment. We added new results with various seed numbers and confidence intervals in Table 2 to justify the performance of our method and revised the caption for clarity.
8. Section 3: added parameters for the density function used in Section 3.2 and Section 3.3.
9. Section 4: emphasized the potential of using URLOST for recovering topology.
10. Section 5: emphasized how other “topology recovering” work cannot be easily integrated with other self-supervised learning to result in an effective method.
11. Appendix A1: added the motivation of using density adjustment for spectral clustering using intuition from function analysis.
12. Appendix A6: we added more experiment and implementation details for better transparency and reproducibility. We will also release the full codebase to aid reproduction when the paper is accepted.

Minor wording changes across the manuscripts are marked in blue.
The current manuscript exceeds the 9-page requirement as parts that will be removed are still visibly struck out for demonstration purposes. These parts will be properly refined in the camera-ready version.

Next, we reply to each reviewer to address the specific questions.

---

### Meta-Review · Area_Chair_jdGE · 2023-12-08

**Metareview:**

The authors present a novel unsupervised learning method meant to overcome limitations of convolutional neural networks, namely that they assume approximate translation invariance and a fixed grid topology for the input structure. The reviewers all found the proposed method interesting, and numerous improvements were made during the review process, however the paper has fundamental flaws and needs to be redone from the ground up. The motivation for the paper seems entirely confused. First of all, the authors use jargon in a way that is not standard and in some cases is simply wrong. They refer to the property that inputs should be treated in a translation-invariance way as "stationarity", but stationarity is a property of time series, not images. They further state that it is a fundamental property of self-supervised learning that inputs must be translation invariant and have fixed topological structure. This is such a fundamental error that I am surprised that none of the reviewers mentioned it. Self-supervised learning is *only an objective function* and can be applied to *any* model architecture. Translation invariance and inability to handle non-grid inputs meanwhile is a fundamental property of *convolutional* neural networks, but it is hardly true of all models. Convolutional neural networks are no longer even the dominant neural network architecture used in vision, having been largely supplanted by Vision Transformers. The motivating reasons given for the development of URLOST are precisely the motivating reason for the development of graph neural networks, a vast and successful branch of machine learning that the authors hardly touch on, including SSL. The paper will have to be reframed in terms of competing with graph neural networks and rewritten with an entirely different motivation and framing.

**Justification For Why Not Higher Score:**

This is addressed in the metareview.

**Justification For Why Not Lower Score:**

This is addressed in the metareview.

---

### Decision · Program_Chairs · 2024-01-16

Reject